# Off-Label and Unlicenced Medicine Use among Hospitalised Children in South Africa: Practice and Policy Implications

**DOI:** 10.3390/pharmacy11060174

**Published:** 2023-11-09

**Authors:** Hlayiseka Mathevula, Natalie Schellack, Samuel Orubu, Brian Godman, Moliehi Matlala

**Affiliations:** 1School of Pharmacy, Sefako Makgatho Health Sciences University, Molotlegi Street, Pretoria 0204, South Africa; brian.godman@strath.ac.uk (B.G.); moliehi.matlala@smu.ac.za (M.M.); 2Department of Pharmacology, University of Pretoria, Pretoria 0084, South Africa; natalie.schallack@up.ac.za; 3Pharmacy Department, Niger Delta University, Yenagoa P.O. BOX 72, Nigeria; samuelorubu@lycos.com; 4Global Strategy Lab, York University, Toronto, ON 4700, Canada; 5Department of Pharmacoepidemiology, Strathclyde Institute of Pharmacy and Biomedical Science (SIPBS), University of Strathclyde, Glasgow G4 0RE, UK

**Keywords:** off-label, unlicensed, paediatrics, evidence-based

## Abstract

Background: Information regarding off-label and unlicensed medicine use among South African children is limited. This is a concern as the prescribing of off-label and unlicensed medicines can lead to issues of effectiveness and safety as well as raise liability issues in the event of adverse events. This potentially exposes physicians to legal penalties. Consequently, we sought to determine the prevalence of off-label and unlicensed medicine use among paediatric patients in South Africa to provide future direction. Methods: This study retrospectively examined the use of medicine in a point-prevalence survey study (PPS) involving paediatric patients aged (0–2 years) admitted to selected public hospitals in Gauteng Province, South Africa. Data were collected per hospital over two days between February 2022 and July 2022. Demographics, duration of treatment, diagnosis, and medicines prescribed were collected from patient medical records using a mobile application. Prescribed medicines were reviewed against the medicine formularies and other databases to assess their appropriateness. Results: From three academic hospitals, 184 patient records were reviewed. A total of 592 medicines were dispensed, of which 379 (64.0%) were licensed and 213 (36.0%) were used off-label/unlicensed for paediatric patients 0–2 years of age. The most prevalent off-label and unlicensed medicines were multivitamins (n = 32, 15.0%) and ampicillin injections (n = 15, 7.0%). Conclusion: The frequency of unlicensed and off-label medicine prescribing shown in this study is consistent with the literature and can be considered high. This practice can pose a risk because it adversely affects patients if not properly regulated. Attention is needed to ensure future high-quality, safe, and effective use of medicines.

## 1. Introduction

Medicine prescribing and the use of medicines in paediatric patient care have been a global issue for a number of years, with high rates of off-label prescribing seen in paediatric patients in a number of studies [1,2,3,4,5,6]. In general, studies indicate that the global level of off-label or unlicensed use of medicines among hospitalised children ranges from 12 to 70% for prescriptions and can reach up to 100% in some studies [7,8]. Recent reviews also suggest that the unlicensed use of medicines can account for up to 75% of medicine use among hospitalised children in some studies [8]. Having said this, Oshikoya et al. (2017) reported off-label prescriptions of only 7.7% among children with chronic diseases attending specialty paediatric clinics in Nigeria [9]. However, the potential for drug–drug interactions was higher among paediatric patients in an earlier study [10]. In addition, there were considerable concerns with the off-label use of pentazocine among paediatric surgical patients in Nigeria, and most children experienced between two and seven adverse events [11]. Developing countries, including South Africa, are acutely affected by off-label and unlicensed use of medicines because people aged between 0 and 18 years constitute an appreciable proportion of the population, and they are more prone to infectious diseases [12,13,14,15]. For instance, Southern Africa has a high proportion of children born to mothers with HIV, which is very different from higher-income countries [16].

The high rates of both off-label and unlicensed use of medicines globally are mainly due to the paucity of clinical trials in children [4,6,17]. The absence of thorough and careful medicine regulatory registration evaluation in paediatrics has also been cited as contributing to off-label prescribing; however, the WHO’s Vigi-Base system is being increasingly used to identify paediatric safety signals [18,19]. The absence of clinical trials in paediatric patients due to economic and ethical concerns further complicates the medicine approval process for this vulnerable population [6,20,21]. Encouragingly, we are now seeing an increase in paediatric biobanks to enhance research, including translational research, for children [22]. Since the recognition of paediatrics as “therapeutic orphans” in the late 1960s and continuing [23,24], there has been global acceptance of the requirement to undertake clinical trials in infants and children to improve their health [25]. Encouragingly, following the promulgation of the Paediatric Research Equity Act of 2003 (PREA) and the Best Pharmaceutical for Children Act (BPCA) of 2003—updated in 2022 to develop age-appropriate medicines—there have been significant changes in paediatric labelling, with these acts addressing previous laws restricting pharmaceutical companies from marketing medicines for children without research data to prove safety for use in children.

Prescribing off-label and unlicensed medicines can be unavoidable when there is no other option, with off-label use regularly included in paediatric guidelines [26]. This includes managing children with tuberculosis (TB), where there are currently no age-appropriate formulations suitable for preventing and treating tuberculosis among the paediatric population in South Africa, despite several such formulations now being commercially available in other parts of the world [27]. Here, the benefit of treating TB using adult formulations appears to outweigh the risks; however, paediatric formulations are preferable, although these are currently unavailable in South Africa [27,28]. However, potential issues with the acceptability/swallowability of adult formulations, their dosing, and their side effects can adversely affect subsequent efficacy and safety.

However, despite changes in these acts, providing high-quality, safe, and efficacious medicine remains a problem for children. This is because paediatric patients cannot be compared with adult patients because their pharmacokinetics and pharmacodynamics change throughout infancy. The continued use of medicines approved only for prescribing in adults and their subsequent use in children leaves considerable uncertainty about their relative efficacy and safety [2,29,30]. Consequently, there is an urgent need for accelerated research and development of age-appropriate medicines to ensure their safe and effective use in paediatric patients.

While such off-label/unlicensed practices are well-characterised in several developed countries and regions, there are concerns about limited studies in developing countries, including South Africa [2,30]. The lack of studies in developing countries is a concern, especially with, for instance, an appreciably greater prevalence of infectious diseases, including HIV and TB, in developing versus developed countries. Consequently, we sought to start addressing this information gap by providing information on the use of medicines in paediatric patients aged 0–2 years of age in the public sector of South Africa. The findings can be used to guide key stakeholder groups in South Africa and other developing countries on suggested ways to improve the management of these young children in South Africa and beyond.

## 2. Methods

### 2.1. Study Design and Setting

This was a retrospective, multicentre, quantitative data review of medicine used in children (0–2 years of age) using a point-prevalence survey (PPS) study approach designed to determine the type and extent of medicines (off-label and unlicensed) prescribed to children (0–2 years) in academic hospitals in South Africa. Initially, the intention was to conduct this research in four academic hospitals in Gauteng Province, South Africa. However, whilst permission was obtained for all four hospitals, access was ultimately granted to only three. Gauteng Province was selected for this initial study because of the ease of access. Academic hospitals were selected because they provide specialised care, e.g., paediatric, neonatal intensive care, oncology, and paediatric surgery. The combined bed capacity of the four conveniently selected academic hospitals for the survey was 927. As per PPS study designs, the total number of beds was used as the population size determinant for this study.

### 2.2. Sample Size and Strategy

Files of paediatric patients aged 0–2 years hospitalised and available in the ward during the study period were included. The required sample size of 234 was calculated using a 95% confidence interval with a 50% proportion and a margin of error of 3%. A systematic sampling strategy was used whereby a patient in the ward was selected for every file of 0–2-year-olds until the sample size was achieved.

### 2.3. Data Collection

The data collected were all the medicines prescribed among paediatric patients aged 0–2 years who were admitted and available in the ward on the day of data collection from 08:00–17:00 h. Data were collected from patient medical records with the aid of a skilled data HM collector. Patient information was recorded on the PPS information sheet, accessible as a mobile application. The following information was collected from the patients’ files: their age, weight, and gender; length of therapy; diagnosis; route of administration (oral, intravenous, inhalation, topical, and rectal); and prescribed medications.

The following definitions were used in the PPS forms [31]:Off Label use: Defined as the administration of a drug/medicine in a manner that differs from that recommended in marketing authorisation with respect to age, dose, frequency of administration, route of administration, formulation, and/or indication. Similarly, an approved medicinal product is a medicinal product prescribed and administered in accordance with its marketing authorisation.Un-licensed use: Refers to the use of a medicinal product that has not been approved for marketing by the country’s medicine regulatory authority (the South African Health Product Regulatory Authority (SAHPRA)) [32].

The categories of off-label use included age, weight, absence of paediatric information, lack of paediatric clinical data, contraindication, route of administration, and formulation/dosage form of administration as stated in the literature insert or official compendium. The categories of unlicensed use included medicines not approved by the national medicine regulator [32].

The dates were deidentified, and all patient identification details (names, identity numbers, and patient file numbers) were not recorded to completely delink the patient from the data. Data were collected from February 2022 to July 2022. This period was chosen as it includes the winter, a flu season for the young and old, with anecdotal evidence suggesting an increase in medicine use during this period.

In terms of developmental differences and medicine use, the children in the PPS forms were categorised into preterm new-born infants (born before 37 weeks of pregnancy), term new-born infants (0–28 days), infants (28 days up to 12 months), and paediatrics (1 year up to 2 years) [33].

### 2.4. Data Analysis

The collected data were extracted from a mobile application and imported into Microsoft Excel. An unbiased arbitrator checked and cleaned the collected data to ensure consistency. The information was then entered and analysed using (IBM) SPSS Version 28.0. Descriptive statistics were used to calculate the frequencies and percentages of patient demographics and for all variables relevant to the study objectives. Medicines were described at different levels of the World Health Organisation’s Anatomical Therapeutic Chemical Code (ATC) classification. Subsequently, the frequency of medicines used off-label and unlicensed was calculated using the WHO’s (ATC) classification [34]. The conditions under which these medicines were prescribed were categorised as per the International Classification of Diseases 10th revision (ICD10) classification system [35]. A medicine is characterised by its distinct active ingredient, often known by its International Non-proprietary Name (INN). Consequently, various formulations with the same active ingredient, such as paracetamol syrup and paracetamol drops, were recorded as identical medicines. Binary logistic regression was used to evaluate the association among demographic variables with off-label/unlicensed medicine use at a 95% CI and *p*-value ≤ 0.05 significance level.

### 2.5. Ethics Approval and Consent to Participate

Ethical clearance (SMUREC/P/128/2020) was obtained from the Sefako Makgatho Health Sciences Research Ethics Committee, University of the Witwatersrand (clearance certificate M210426), and the National Department of Health (GP_202011_0470). The study was conducted after obtaining official permission from the hospital administration. No consent to participate was required because this study was a retrospective data review of patient records with no direct contact with the patient, and the data were delinked from the patient’s details. This is similar to other PPS studies in South Africa [36,37,38,39,40].

## 3. Results

After receiving ethical approval from all the potential academic hospitals, access to data was denied by the clinicians at one of the four academic hospitals as they feared that the study findings might be used as evidence during litigation despite the anonymity of the data. Overall, 184 (78.6%) of the envisaged 234 samples of paediatric patient files were subsequently accessed from three of the four academic hospitals.

### 3.1. Demographics

Among the 184 paediatric patient files that were reviewed, 592 medicines were prescribed, translating to an average of 3 medicines per patient file. A total of 79.3% of the files were for children between the ages of 0 and 1, as indicated in the study population (Table 1). The mean weight of the study population was 4.62 kg (SD ± 3.63 kg), with a little more than half (51.6%) of the patients being male and the majority, 96.7%, African.

### 3.2. Prevalence of Off-Label/Unlicensed Use in Children (0–2 Years)

The prevalence of off-label or unlicensed medicine use in children aged 0–2 years included in this study was 36% (213/592). Overall, 177 medicines (29.9%, n = 592) were prescribed off-label, while 36 (6.1%, n = 592) were categorised as unlicensed. Off-label prescribing was most prevalent among neonates aged 0–28 days, with 84 cases (39.4% of the 213 cases). On the other hand, unlicensed medicine use was most prevalent among infants aged between 29 days and 1 year, with 17 cases (8.0% of the 213 cases), as illustrated in Figure 1.

### 3.3. The Top 10 Most Prescribed Medicines at ATC Level 5, Chemical Substance, or INN

Six of the top ten most prescribed medicines were off-label or unlicensed (Table 2).

It is worth noting that the use of off-label and unlicensed medicines was not mutually exclusive. In fact, every single paediatric patient (n = 184) included in this study received at least one off-label or unlicensed medicine during their hospital stay. The most frequently prescribed off-label medicine was intravenous caffeine (5.2%, n = 31 of the 592). On the other hand, the medicine that was predominantly used in the unlicensed category was multivitamin syrup/drops, accounting for 6.4% of the cases.

### 3.4. Off-Label Use by Age Group

Off-label medicine use varied by age group. Table 3 provides an overview of the top 10 off-label medicines stratified by age group.

In children aged 0–28 days, ampicillin injections were the most frequently prescribed off-label medicine, with a total of 15 cases, accounting for 7.0% of all medicines (n = 512). Caffeine citrate injections were the second most prevalent off-label medicine, with 11 cases (5.2%). Additionally, gentamicin injections were identified as a prevalent off-label medicine exclusively among the 0–28-day age group, with a total of nine cases (4.3%).

Injections were the most common (80%) dosage form of off-label medicines used in this age group. In contrast, oral dosage forms—tablets, capsules, and syrups—were most common (90%) in infants. Consequently, most medicines were not established for use in neonates or manipulated in infants.

### 3.5. Unlicensed Medicine Use by Age Group at the ATC Level 4 Chemical Subgroup

The most used unlicensed medicine across all ages was multivitamins (A11), and probiotics (A07) were the medicines mostly used unlicensed in infants aged from 29 days to two years. In children aged 0–28 days, multivitamins were mostly used unlicensed, with 12 cases (5.8% of 213). Again, in children aged 29 days to one year, multivitamins were the most common unlicensed medicine with 11 cases (5.3% of 213), followed by probiotics in 2 cases (0.9% of 213). Lastly, in children aged 1 to 2 years, multivitamins were the most common unlicensed medicine with five cases (2.3% of 213), followed by probiotics with two cases (0.9% of 213), as indicated in Table 4.

### 3.6. Conditions Associated with Off-Label and Unlicensed Medicines by ICD-10 Codes

The data presented in Table 5 highlight the most common medical conditions, as indicated by their corresponding ICD-10 codes, in which medicines were used off-label or unlicensed. These were identified based on their frequency within the dataset and comprised 81.7% of diagnosed medical conditions in the sample with off-label or unlicensed medicine use.

Bacterial infections (ICD-10 code A49.9) were the most prevalent, accounting for 26.8% of the cases. Within this category, the majority of cases were prescribed off-label or unlicensed medicines in all age groups, with the highest proportion in the 0–28-day age group (61.4%).

### 3.7. Off-Label and Unlicensed Medicine Use by Therapeutic (ATC Level 2) and Age Groups

The most common (81.8%) off-label and unlicensed medicines are shown in Table 6.

Anti-bacterial for systemic use (ATC code J01), used off-label, was the most prevalent medicine category, comprising 26.3% (56/213) of all off-label and unlicensed medicines (n = 213). The off-label use of this category was highest in the 0–28-day age group (62.5%). The next most frequent medicine group, accounting for 17.4% (37 cases) of all off-label/unlicensed medicines, were vitamins (ATC code A11). Vitamins were unlicensed and mostly prescribed to infants aged 29 days–1 year.

No statistically significant associations were found between patient demographics/health-related variables and off-label/unlicensed use using binary logistic regression analysis. The Chi-Square Tests between the reason for off-label/unlicensed use and patient demographics (age categories and patient weight) were found to be statistically significant with a *p*-value of <0.001.

## 4. Discussion

To the best of our knowledge, this is the first study to report the prevalence of off-label use and unlicensed medicine use among the paediatric population aged zero to two years in South Africa. The frequency of unlicensed and off-label medicine prescribing in our study is consistent with some of the published literature and can be considered high [21,32,33,41,42,43,44,45,46,47,48,49,50,51]. We have seen in published studies among LMICs that off-label use among hospitalised paediatric patients can account for up to 99.5% or more of prescriptions [4,6,7,8,15,35,43,50,51,52,53,54]. However, whilst the percentage of off-label prescribing in our study was appreciably lower than the rates seen in a number of LMICs at 36% for off-label/unlicensed use, this does not negate potential concerns in a number of these very young infants. This practice can pose a risk because it can adversely affect young infants if, for instance, doses and their implications are not properly regulated in hospitals [4,6,7]. Very young infants are more susceptible to side effects when prescribed off-label medicines due to differences in pharmacokinetics and pharmacodynamics compared to adults [43]. This prescribing trend is particularly prevalent in infants aged 0–28 days, with 44% of infants in our study receiving off-label or unlicensed medicines. This rate is higher than in older infants and is consistent with global studies [54,55,56]. The heightened risk stems from the fact that neonates’ renal and hepatic functions are not fully developed [13,21,50,53,56,57,58]. Furthermore, there is a limited evidence base for the safety and efficacy of these medicines in very young infants [35,59,60,61,62].

We saw an average of three medicines prescribed per infant in our study, similar to other studies [31,53,63]. However, this was much higher than in Norway (0.8) and Spain (1.5) [64], and slightly lower than in Italy (3.7) [65], Malaysia (where the median number was 4 [63]), and Indonesia (where the median number of medicines prescribed was 9 [54]). Overall, paediatric patients are exposed to a high number of off-label and unlicensed medicines, which could lead to suboptimal clinical efficacy and unanticipated side effects. This needs to be addressed going forward, enhanced by regularly reviewing the evidence base for their use [66]. This is because for many medicines typically prescribed in neonatal and young infant ICUs, safety and efficacy data for neonatal pharmacotherapy are lacking, with an appreciable number of neonates in ICUs being prescribed medicines that are not approved or are used off-label [4,7,43,67]. Neonatal pharmacotherapy and prescribing practices require special attention, primarily because, as mentioned, neonates have unique pharmacokinetic and pharmacodynamic profiles compared to older children and adults. These differences can influence drug absorption, distribution, metabolism, and elimination [68,69]. Consequently, the efficacy and safety of medications in neonates can vary significantly from other populations [29,70]. Accurate dosing, vigilant monitoring, and a comprehensive understanding of the drug’s effects are essential to prevent potential adverse reactions [71] and to ensure therapeutic efficacy in this very young population. Consequently, healthcare professionals must remain updated on the latest research, guidelines, and recommendations related to neonatal pharmacotherapy [17,69,72]. This is very important for tertiary hospitals in the public healthcare system in South Africa and beyond going forward.

In our study, similar to others [13,35,50,73,74,75,76], systemic antibiotics were the most frequently prescribed medicines, especially among neonates. This high use of antibiotics reflects the fact that the top indication for off-label or unlicensed use in medicines in our study was for bacterial infections at 26.8%, higher than studies in Spain at 12.0% [42], France at 22% [45], and Uganda at 18.9% [77]. However, this study found lower use compared to studies conducted in the Western Cape, South Africa, which had a rate of 39% [78], and Jordan, where the rate of bacterial infections was 54.1% [79]. This high rate of antibiotic prescribing is perhaps not surprising, as academic hospitals typically treat more premature children with low birth weight and sepsis than secondary or community hospitals [80]. However, it is important to fully monitor the prescribing of antibiotics in this population because sepsis is the leading cause of neonatal death globally, killing more than 1 million neonates worldwide each year, with appreciably higher mortality rates in LMICs [81,82,83]. This results in antibiotics being among the most commonly prescribed drugs in neonatal intensive care units [84,85,86]. Proper dosing of antibiotics is critical as under- or over-dosing can increase antimicrobial resistance (AMR) [56,87], which is a concern with mortality from AMR growing globally, with the highest mortality rates from AMR currently seen in sub-Saharan Africa [88]. In addition, an estimated 31.0% of neonatal sepsis deaths are currently due to AMR and are rising [89]. In addition, under- or over-dosing medicines (including antibiotics), which may result from their off-label or unlicensed use, is a concern, as this poses a risk of reduced effectiveness, increased adverse reactions, or both, along with potentially increasing AMR [75,87]. We have seen antimicrobial stewardship (ASP) programmes effectively introduced in hospitals across Africa in recent years to improve antimicrobial prescribing, with hospital pharmacists playing a key role [90,91,92,93]. These ASP exemplars should provide guidance to key stakeholder groups in South Africa and beyond to address the inappropriate use of antibiotics alongside rising AMR on the continent.

There was also appreciable prescribing of caffeine in our study. Thomas (2014) classified caffeine as unlicensed in all dosage formulations [49]. However, at the time of our study, caffeine was registered with the regulators in South Africa as an injectable but was prescribed and administered orally. It is worth noting that there is currently no commercially available oral solution specifically formulated for caffeine in South Africa. Nevertheless, the Department of Health strongly recommends the oral route of administration for caffeine as per EML guidance [94,95,96,97,98,99]. Vitamins were also the most common medicine used unlicensed or unapproved in our study. This was due to a lack of marketing authorisation for vitamins from the regulatory body in South Africa. Currently, the South African Health Products Regulatory Authority classifies vitamins as dietary supplements. Previously, they were considered food/dietary supplements and did not undergo the same registration process as conventional medicines. Consequently, this might be the justification for unlicensed medicine status. This again needs to be looked at in light of their considerable use among this patient population in South Africa.

Age, route of administration, and dosage were the most common reasons for off-label prescribing in our study, similar to previous studies [7,32,49,100]. Due to the lack of suitable oral drug forms for neonates and young children, tablet splitting and dissolution in sterile liquids before administration are common, as seen in other studies [101]. However, both within and outside hospitals, the practice of tablet splitting or dissolution can place nurses and caregivers in the difficult situation of having to prepare and administer the medications according to current recommendations [102]. Furthermore, child acceptance of these manipulated medicines could be compromised [103]. It is also difficult for physicians to adjust dosages over time to ensure the adequate safety and efficacy of the prescribed medicines because the predominant method of manipulation is mixing with liquid and food. Alongside this, for certain medicines, food–drug interactions can appreciably affect their bioavailability and therapeutic efficacy, which needs to be considered when administering them [101,102]. Drug delivery and uptake can also be influenced by the medicine’s swallowability, taste, smell, texture, and appearance, which can be altered when adjusting or diluting doses [103,104], potentially leading to worse outcomes as a result. Consequently, this again needs careful monitoring.

The role of Drug and Therapeutic Committees (DTCs) in promoting rational and evidence-based prescribing practices is also pivotal, especially in addressing the current lack of rigorous regulatory evaluation of paediatric formulations [105]. We have seen the role of DTCs grow in South Africa compared with other African countries; however, more needs to be accomplished going forward [106,107,108,109,110]. Standard Operating Procedures (SOPs) can aid DTCs in streamlining and regulating inappropriate medication use, particularly concerning dosing for young infants, and preventing adverse drug reactions [110]. Monitoring adherence to these SOPs elevates the quality of care and strengthens the feedback loop, which is essential for continuous improvement.

Collaboration between DTCs and regulatory authorities, including SAHPRA, is paramount to bolstering the impact of these endeavours. Such a partnership would ensure up-to-date information on drug safety, efficacy, and quality for paediatric populations. By prioritising clinical trials tailored to the paediatric population and addressing issues related to off-label and unlicensed medication use, a comprehensive understanding of drug effects and risks can be established for this vulnerable group. Ultimately, by harnessing the synergy of DTCs and SAHPRA, South Africa has the potential to be a beacon for evidence-based paediatric medicine administration, extending best practices across the African continent. This is similar to the situation seen with the implementation of national action plans to reduce AMR across Africa, with ongoing activities in South Africa being more advanced than those seen in a number of other African countries and providing direction [90,105,110,111,112,113,114,115,116,117].

We are aware of a number of limitations with this study. Firstly, as this was a PPS study design, we only recorded medicines prescribed that day. Consequently, we were unable to link any off-label or unlicensed medicine with any contribution to any adverse drug reaction. Furthermore, this pilot study was conducted in only one province. However, despite these limitations, we believe the findings are robust.

## 5. Conclusions

In this study, off-label and unlicensed medicine use appear very common among paediatric patients aged 0 to 2 years of age admitted to public, academic hospitals in South Africa, similar to other studies. Addressing the issue of off-label and unlicensed drug use in paediatric patients is vital for promoting patient safety and improving healthcare outcomes going forward. In the first instance, this includes a greater role for DTCs in hospitals to develop appropriate standards and monitor their implementation, as well as encouraging greater understanding of neonatal pharmacology. In addition, seeking to instigate clinical trials, where possible, tailored to the paediatric population should be initiated. We will be following up on these suggestions in the future.

## Figures and Tables

**Figure 1 pharmacy-11-00174-f001:**
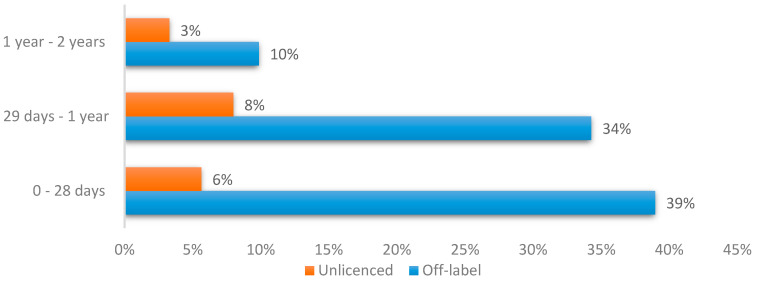
Prevalence of off-label and unlicensed medicine use in different age categories within a sample of 184 children aged 0–2 years old across three academic hospitals located in Gauteng Province, South Africa.

**Table 1 pharmacy-11-00174-t001:** Patient demographic data.

Age Categories		
Age	Prevalence	Percentage
0–28 days (neonates)	70	38.0%
29–1 year (infants)	76	41.3%
1–2 years (paediatrics)	38	20.7%
Mean age	6.2 months	
SD±	±236.6	
**Total**	**184**	**100.0%**
**Gender**		
**Gender**	**Prevalence**	**Percentage**
Male	95	51.6%
Female	89	48.4%
**Total**	**184**	**100.0%**
**Weight categories in grams**		
**Weight**	**Prevalence**	**Percentage**
630–5000 g	116	63.0%
5001–10,000 g	58	31.5%
10,001–15,000 g	7	3.8%
15,001–20,000 g	3	1.6%
Mean	4625.9	
SD±	3633.7	

**Table 2 pharmacy-11-00174-t002:** Top 10 prescribed medicines by chemical substance (ATC Level 5).

ATC Code	Medicine Name	Dosage Form	Freq. N = 592	Percentage	Used On/Off-Label/Unlicensed
A11CC	Vitamin D	Drops	38	6.4	on-label
B03BB	Folate	Syrup	36	6.1	on-label
A11AB	Multivitamin	Syrup/drops	32	5.4	unlicensed
N06BC01	Caffeine 20 mg per mL	Injection	31	5.2	off-label
J01CA01	Ampicillin	Injection	27	4.6	off-label
B03AA03	Ferrous Gluconate	Syrup	25	4.2	on-label
N02BE01	Paracetamol 120 mg per 5 mL	Syrup	22	3.7	off-label
J05AG01	Nevirapine solution 50 mg per 5 mL	Solution	20	3.4	on-label
J01GB06	Amikacin	Injection	18	3.0	off-label
J01CR02	Amoxicillin 500 mg/Clavulanic acid 100 mg per 20 mL	Injection	14	2.4	off-label

**Table 3 pharmacy-11-00174-t003:** Categories of the top 10 off-label medicines (ATC Level 5) in children under 2 years of age.

ATC Code	Medicine	Dosage Form	Freq. N = 213 (%)	Reason for Being Off-Label
0–28 days				
J01CA01	Ampicillin	Injection	15 (7.0)	Not established for use
N06BC01	Caffeine 20 mg per mL	Injection	11 (5.2)	Administration orally
J01GB03	Gentamycin	Injection	9 (4.3)	Not established for use
N02BE01	Paracetamol 120 mg	Syrup	5 (2.3)	Not established for use (<2 months)
A03FA01	Metoclopramide 2mg/mL	Injection	4(1.9)	Not established for use (<1 year)
N02BE01	Paracetamol 10 mg/mL	injection	4 (1.9)	Not established for use
J01DH02	Meropenem 500 mg	injection	3 (1.4)	Not established for use (<3 months)
N05CF08	Midazolam 50 mg	injection	2 (1.0)	
A11HA02	Pyridoxine 25 mg	Tablet	2 (1.0)	Dosage form manipulation
J01CR02	Amoxicillin 500 mg/clavulanic acid 100 mg/20 mL	Injection	2 (1.0)	High dose/weight of patient
ATC Code	Medicine	Dosage form	Freq. N = 213 (%)	Reason for being off-label
29 days to 1 year				
N02AB03	Fentanyl	injection	5 (2.5)	Not established for use (<2 years)
J01DH02	Meropenem 500 mg	Injection	5 (2.5)	Not established for use (<3 months)
J01CR02	Amoxicillin 500 mg/Clavulanic acid 100 mg/20 mL	Injection	5 (2.5)	High dose/weight of patient
	Isoniazid 100 mg	Tablet	5 (2.5)	Dosage form manipulation
A03FA01	Metoclopramide 5 mg per 5 mL	Syrup	4 (1.9)	Not established for use (<1 year)
N06BC01	Caffeine 20 mg per mL	Injection	4 (1.9)	Route of administration (orally)
G01AA03	Amphotericin B	Injection	4(1.9)	Not established for use (manufacturer)
N07BC02	Methadone 10 mg/mL	solution	3 (1.4)	Not established for use in children
A12BA01	Potassium chloride	Tablet	3 (1.4)	Dosage form manipulation
N05BA01	Diazepam	Tablet	3 (1.4)	Dosage form manipulation
ATC Code	Medicine	Dosage form	Freq. N = 213 (%)	Reason for being off-label
1 year to 2 years				
C03AA03	Hydrochlorothiazide	Tablet	7 (3.3)	Dosage form manipulation
J04AC01	Isoniazid 100 mg	Tablet	5 (2.3)	Dosage form manipulation
J01CR02	Amoxicillin 500 mg/Clavulanic acid 100 mg/20 mL	Injection	2 (1.0)	High dose/weight of patient
A12BA01	Potassium chloride	Tablet	2 (1.0)	Dosage form manipulation
R06AE07	Cetirizine	Syrup	1(0.5)	Not established for use (<2 years)
A04AA01	Ondansetron 4 mg	Tablet	1 (0.5)	Not established for use (<4 years)/dosage form manipulation
G04BD04	Oxybutynin 5 mg	Tablet	1 (0.5)	Not established for use (<5 years)/dosage form manipulation
J04AK01	Pyrazinamide	Tablet	1 (0.5)	Dosage form manipulation
J04AM02	Rifampicin/Isoniazid	Tablet	1 (0.5)	Dosage form manipulation
A02BC01	Omeprazole 10 mg	Capsule	1 (0.5)	Dosage form manipulation

**Table 4 pharmacy-11-00174-t004:** Most prevalent unlicensed medicines used.

ATC Code	Medicine	Dosage Form	Freq. N = 213(%)	Reason for Being Unlicensed
0–28 days				
A11AB	Multivitamin	Syrup	4 (2.0)	Not approved by SAHPRA
A11AB	Abidec multivitamin	Drops	8 (3.8)	Not approved by SAHPRA
29 days to 1 year				
A11AB	Abidec multivitamin	Drops	7 (3.3)	Not approved by SAHPRA
A11AB	Multivitamin	Syrup	4 (2.0)	Not approved by SAHPRA
A07FA01	Probiotics	Drops	2 (0.9)	Not registered with SAHPRA
1 year to 2 years				
A11AB	Multivitamin	Syrup	5 (2.3.0)	Not approved by SAHPRA
A07FA01	Probiotics	Drops	2 (0.9)	Not registered with SAHPRA

**Table 5 pharmacy-11-00174-t005:** Most common medical diagnoses or conditions per ICD-10 codes for which medicines were prescribed off-label or unlicensed in three different age groups.

ICD10 Code	Description	Freq. N = 213	%	0–28 Days	29 Days–1 Year	1–2 Years
A49.9	Bacterial infections	57	26.8	35 (61.4%)	15 (26.3%)	7 (12.3%)
E56	Vitamin deficiency, unspecified	32	15.0	12 (37.5%)	15 (46.9%)	5 (15.6%)
R52	Pain, not elsewhere classified	21	9.9	13 (61.9%)	7 (33.3%)	1 (4.8%)
P28.3	Primary apnoea of new-born, unspecified	15	7.0	11 (73.3%)	4 (26.7)	0 (0.0%)
A15	Respiratory tuberculosis	13	6.1	2 (15.4%)	8 (61.5%)	3 (23.1%)
R11	Nausea and vomiting	11	5.2	4 (36.4%)	6 (54.5%)	1 (9.1%)
R60.9	Oedema	7	3.3	6 (85.7%)	1 (14.3%)	0 (0.0%)
F13.20	Sedative, hypnotic, or anxiolytic	7	3.3	2 (28.6%)	5 (71.4%)	0 (0.0%)
Y40.7	Fungal infections	6	2.8	1 (16.7%)	5 (83.3%)	0 (0.0%)
E87.6	Potassium deficiency	5	2.3	0 (0.0%)	3 (60.0%)	2 (40.0%)

**Table 6 pharmacy-11-00174-t006:** Categories of off-label or unlicensed medicines used by age group (ATC Level 2).

Top 10 Used ATC Code of Medicine Categories Used Off-Label/Unlicensed							
ATC Code	Therapeutic Subgroup	Off-Label/Unlicensed	Freq. N = 213	%	0–28 Days	29 Days–1 Year	1–2 Years
J01	Anti-bacterial for systemic use	Off-label	56	26.3	35 (62.5%)	14 (25.0%)	7 (12.5%)
A11	Vitamins	Unlicensed	37	17.4	14 (37.8%)	17 (45.9%)	6 (16.2%)
N02	Analgesics	Off-label	18	7.0	11 (61.1%)	7 (38.9%)	0
N06	Psychoanaleptics	Off-label	15	7.0	11 (73.3%)	4 (26.7%)	0
J04	Anti-mycobacterial	Off-label	13	6.1	2 (15.4%)	8 (61.5%)	3 (23.1%)
A03	Drugs for functional gastrointestinal disorders	Off-label/unlicenced	10	4.7	4 (40.0%)	6 (60.0%)	0
C03	Diuretics	Off-label	9	4.2	6 (66.7%)	3 (33.3%)	0
N05	Psycholeptics	Off-label	7	3.3	2 (28.6%)	5 (71.4%)	0
J02	Antimycotics for systemic use	Off-label	6	2.8	1 (16.7%)	5 (83.3%)	0
N01	Anaesthetics	Off-label	5	2.3	2 (40.0%)	3 (60.0%)	0

## Data Availability

The datasets generated or analysed as part of this study are included in this published paper.

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
