# Peer review of "Off-Label and Unlicenced Medicine Use among Hospitalised Children in South Africa: Practice and Policy Implications"

_pharmacy, 2023, doi:10.3390/pharmacy11060174_

Round 1

Reviewer 1 Report

Comments and Suggestions for Authors

Please refer to the attached file for the comments

Comments on the Quality of English Language

The English language can be improved.

Author Response

Reviewer 1

The study conducted by Mathevula H et.al. reported the prevalence of off-label and unlicensed medicine use in pediatric patients in South African hospitals, highlighting potential safety and regulatory concerns. The study highlights the need to improve the safety of medication use in pediatric patients and inform healthcare professionals and regulators about potential risks associated with off-label and unlicensed medicine prescribing. The following minor points need to be considered by the authors.

In the Introduction section, paragraph no.3 Lines 66-69, It is better to merge this information with the first paragraph where authors have already discussed epidemiological data of off-label medicines use.

Author comment: thanks for your remarks the two paragraphs have been merged.

In the Introduction section, paragraph no.4 Line 71-72, This information should be merged with the second paragraph where authors discussed the absence of the regulations that leads to off-label use, because these statements prove the same point and can be combined.

Author comment: thanks for your remarks the section is now merged with paragraph 2.

In the Introduction section paragraph no. 5 Lines 79-85, This information should be merged with the information in paragraph no.2 both are seemingly talking about the clinical trial and how some act reforms are being implemented.

Author comment: thanks for your remarks the information has been merged to the second paragraph.

In the introduction section, paragraph no.6 Lines 87-90, This information is better in paragraph seven of the introduction, you should talk about the difference between children's and adults' pharmacodynamics and pharmacokinetics and try to merge these lines with the paragraph number 7 stating only the necessary problems and the needs that lead to conduct this study.

Author comment: thanks for your remarks the section has been corrected.

In the Method section, study design and settings, Lines 108-114, Kindly combine the paragraph with the first paragraph of study design and settings. It should be the continuation.

Author comment: thanks for your remarks the two paragraphs have been combined.

In the method section, Data analysis, Line 154-169, Kindly combine all the paragraphs into a single paragraph.

Author comment: thanks for your remarks all paragraphs have been combined.

In the result section, Line 178, Kindly do the double spacing for the result section, there is no space after ethical approval.

Author comment: thanks for your remarks the section has been corrected.

In the result section, Table no.2, Kindly write "Percentage" on the 5th column of table no.2.

Author comment: thanks for your remarks the section has been corrected.

In the result section, Lines 230-232, Kindly write more descriptively with the total number and percentages of the vitamins and probiotics. You can report the one with the highest percentages across the age group.

Author comment: thanks for your remarks the section has been corrected.

In the discussion section, first paragraph, Line 269, You have specified that the frequency of unlicensed and off-label medicine prescribing is consistent with existing literature, but none of the references has been cited. I suggest citing the studies that are consistent with your findings.

Author comment: thanks for your remarks the section has been corrected.

In the discussion section, first paragraph, Line 275-283, Kindly, concise the paragraph, remove the repetitive information, and only report why very young infants have more risk of side effects and why there is little evidence related to studies globally that you have cited.

Author comment: thanks for your remarks the section has been corrected.

In the discussion, first paragraph, Lines 277-279 you state that, there is a high prevalence of off-label medication in very young infants, but did not cite any study kindly provide the citation.

Author comment: thanks for your remarks the section has been corrected.

In the discussion section, second paragraph, line 286-287, Kindly include studies from other countries as well, do you find it to be lower or higher than in order countries or not? (other than the Indonesia) The evidence you provided is very limited.

Author comment: thanks for your remarks the section has been corrected.

In the discussion section, second paragraph, Lines 293-296, it is the repetition, kindly just add a comment about the need to consider neonatal pharmacotherapy and prescribing practices with relevant citations.

Author comment: thanks for your remarks the section has been corrected.

In the discussion section, third paragraph, Line 299-301, Kindly include more evidence from other studies as well that have found different results (higher or lower) incidents of bacterial infection and also the use of antibiotics, reporting only bacterial infection does not provide the full context also include the prevalence of antibiotic use you found in your study along with the bacterial infection incidents.

Author comment: thanks for your remarks the section has been corrected.

In the discussion, section the 4 and 5 paragraphs, Lines 317 – 335 should be combined removing unnecessary information, specifically about the practise of Caffeine use in South Africa, Kindly be precise and relate your finindings to the regulatory guidelines.

Author comment: thanks for your remarks the section has been corrected.

In the discussion section, paragraph no. 7 and 8 Lines 351-376, Kindly summarise the DTC procedure, there is no need to report how it works instead report how SOPs can help in regulating inappropriate medication use. Furthermore, combining these two paragraphs, you can recommend that DTC should collaborate with SAHPRA because it can better monitor the medication use, kindly be very precise with the sentencing it seems repetitive.

Author comment: thanks for your remarks the section has been corrected.

Concluding remarks: this is an interesting study which is very neededful research. However, in my opinion the paper has shortcomings, e.g. it does not consistently follow a scientific writing style and the statements are repetitive and sometimes vague especially in the discussion part. There is excessive number of tables presented in the manuscript that took a lot of space, some tables can be added as supplementary material. In general, the research is well conducted and presented.

Comments on the Quality of English Language: can be improved

Author comment: Thank you. We have now been through the paper with the help of one of the co-authors, a native English speaker with over 500 publications to his name. We trust this is now fine.

Reviewer 2 Report

Comments and Suggestions for Authors

This paper addresses the issue of the possibility and frequency of off-label use of medications in a group of pediatric patients. This problem has been known for many years and currently, apart from the therapeutic difficulties associated with it, is not of significant interest to the scientific community. The advantage of this work is the description of the problem in a developing country with a lower national income per capita and smaller opportunities to deal with it than in Europe. However, the results are not surprising and do not provide any new content.

Comments on the Quality of English Language

no comments

Author Response

Reviewer’s comments:

Reviewer 2

This paper addresses the issue of the possibility and frequency of off-label use of medications in a group of paediatric patients. This problem has been known for many years and currently, apart from the therapeutic difficulties associated with it, is not of significant interest to the scientific community. The advantage of this work is the description of the problem in a developing country with a lower national income per capita and smaller opportunities to deal with it than in Europe. However, the results are not surprising and do not provide any new content.

Author comment: Thank you for your feedback on our study. While we recognise that the issue of off-label use of medications in paediatric patients has been known for years, the context and implications can be very different across nations, especially between developed and developing countries. This is a real concern especially with the high prevalence of both infectious and non-infectious diseases across Africa as well as combined co-morbidities. Consequently, understanding the prevalence of off-label or unlicensed use of medicines is paramount for countries such as South Africa to help improve future medicine use/ patient outcomes. As such, we believe our national backdrop, coupled with unique socio-economic challenges, necessitates such research in South Africa. Overall, we believe the insights derived are crucial to providing context-specific data, which we believe will be of interest to other similar countries. We hope you now agree with us..

Our study aimed not merely to describe the prevalence but also to understand its implications in the South African context and, subsequently, offer recommendations. While the core issue might be universal, we believe the nuances and specifics of its manifestation and potential solutions can vary significantly based on a nation's healthcare infrastructure, economy, and policy framework especially among LMICs. Overall, we believe our findings holds significant importance in aiding policymakers across LMICs tackle similar issues. By shedding light on the current scenario, we aim to provide them with the necessary data to make informed decisions and possibly adjust medical guidelines and policies to better cater to our paediatric population. We appreciate the global perspective on this issue. Still, we believe it is essential to emphasise the need for region-specific research, significantly when it aids the decision-making process in developing nations like ours.

Comments on the Quality of English Language: no comment

Reviewer 3 Report

Comments and Suggestions for Authors

Dear authors,

I have read the manuscript pharmacy-2638584 thoroughly. The submitted manuscript presents the results of a study, considering the off-label use of medicines and the use of unlicensed medicines among the children under the age of 3 in South African hospitals. Overall, the quality of the study and the interest for the readers are approximately average for the manuscripts submitted to this magazine.

Abstract gives all the necessary information about the article. No additional change is needed. Keywords are appropriately chosen.

“Methods” include all the necessary data. No improvement is needed in this part of a manuscript. The same applies to the “Results and Discussion” part. An extensive range of literature sources provides the reader with easy access to relevant information. Self-citation is hardly noticeable.

In my opinion, only small modification in introduction is necessary. In Lines 50–52, the numerical values are too specific. I would suggest: “ranges from 12 to 70 %for prescriptions and may reach up to 100 % in some studies” in Line 50. In line 51, I would suggest that the value of 75 % is written and in the same line there is a mistake: medicines can not be hospitalised.

I would also suggest correcting the first sentence in Lines 164–165. It may be misleading.

Best regards,

Comments on the Quality of English Language

The quality of English language is high. I noticed a few typing errors, so I suggest that you read the whole manuscript again carefully while correcting the paper and also correct it from a linguistic point of view.

Author Response

Reviewer’s comments:

Reviewer 3

Dear authors,

I have read the manuscript pharmacy-2638584 thoroughly. The submitted manuscript presents the results of a study, considering the off-label use of medicines and the use of unlicensed medicines among the children under the age of 3 in South African hospitals. Overall, the quality of the study and the interest for the readers are approximately average for the manuscripts submitted to this magazine. 

Author comment: Thank you.

Abstract gives all the necessary information about the article. No additional change is needed. Keywords are appropriately chosen.

Author comment: Thank you.

“Methods” include all the necessary data. No improvement is needed in this part of a manuscript. The same applies to the “Results and Discussion” part. An extensive range of literature sources gives the reader easy access to relevant information. Self-citation is hardly noticeable.

Author comment: Thank you.

In my opinion, only small modification in the introduction is necessary. In Lines 50–52, the numerical values are too specific. I would suggest: “ranges from 12 to 70 %for prescriptions and may reach up to 100 % in some studies” in Line 50. In line 51, I would suggest that the value of 75 % is written and in the same line there is a mistake: medicines can not be hospitalized.

Author comment: thanks for your remarks the section has been corrected.

I would also suggest correcting the first sentence in Lines 164–165. It may be misleading.

Author comment: thanks for your remarks the section has been corrected.

Comments on the Quality of English Language: The quality of English language is high. I noticed a few typing errors, so I suggest that you read the whole manuscript again carefully while correcting the paper and also correct it from a linguistic point of view.

Author comment: Thank you. We have now been through the paper with the help of one of the co-authors, a native English speaker with over 500 publications to his name. We trust this is now fine.

Round 2

Reviewer 2 Report

Comments and Suggestions for Authors

After the authors introduced changes to the manuscript and took into account the arguments supporting publication, I believe that this manuscript can be published